# Clinical Strategies for Enhancing the Efficacy of CAR T-Cell Therapy for Hematological Malignancies

**DOI:** 10.3390/cancers14184452

**Published:** 2022-09-14

**Authors:** Qianzhen Liu, Zengping Liu, Rongxue Wan, Wenhua Huang

**Affiliations:** 1School of Basic Medical Sciences, Guangdong Medical University, Dongguan 523000, China; 2Department of Pathology, The First People’s Hospital of Foshan, Foshan 528000, China; 3Orthopaedic Center, Affiliated Hospital of Guangdong Medical University, Guangdong Medical University, Zhanjiang 524000, China; 4Guangdong Engineering Research Center for Translation of Medical 3D Printing Application, Guangdong Provincial Key Laboratory of Medical Biomechanics, School of Basic Medical Sciences, Southern Medical University, Guangzhou 510515, China

**Keywords:** immunotherapy, CAR T-cell, efficacy, clinical strategies, acute lymphoblastic leukemia, non-Hodgkin’s lymphoma

## Abstract

**Simple Summary:**

Cellular immunotherapy has achieved great progress in the treatment of hematological malignancies, and cell therapy may become the main force in the treatment of refractory and relapsed malignant tumors in the future. The efficacy of CAR T cells as a potential cell therapy in the treatment of hematological tumors is acknowledged. Given that clinical application strategies for optimizing CAR T-cell therapy are still immature, this article reviews the impact of various application strategies on CAR T-cell therapy.

**Abstract:**

Chimeric antigen receptor (CAR) T cells have been successfully used for hematological malignancies, especially for relapsed/refractory B-cell acute lymphoblastic leukemia and non-Hodgkin’s lymphoma. Patients who have undergone conventional chemo-immunotherapy and have relapsed can achieve complete remission for several months with the infusion of CAR T-cells. However, side effects and short duration of response are still major barriers to further CAR T-cell therapy. To improve the efficacy, multiple targets, the discovery of new target antigens, and CAR T-cell optimization have been extensively studied. Nevertheless, the fact that the determination of the efficacy of CAR T-cell therapy is inseparable from the discussion of clinical application strategies has rarely been discussed. In this review, we will discuss some clinical application strategies, including lymphodepletion regimens, dosing strategies, combination treatment, and side effect management, which are closely related to augmenting and maximizing the efficacy of CAR T-cell therapy.

## 1. Introduction

Cellular immunotherapy of hematological malignancy has achieved remarkable progress in the last decade, especially that based on chimeric antigen receptor (CAR) T cells. CAR T-cells have experienced four generations (Figure 1). First-generation CAR T-cells were found to lack persistence and expansion in pre-clinical studies, while second-generation cells overcame these limitations by containing intracellular costimulatory domain (e.g., CD28 or 4-1BB), later fused to CD3zeta. Third-generation cells with two costimulatory domains of CD28 and 4-1BB showed promising results in persistence and anti-tumor activity [1], while fourth-generation CAR T-cells additionally achieved specific cytokine release through gene editing following T cell activation [2]. CD19-directed CAR T-cells as representative second-generation CAR T-cells have led to unprecedented development in the treatment of B-cell hematological malignancies, with over 90% overall survival [3,4]. It is worth mentioning that Melenhorst et al. of the University of Pennsylvania found that after infusion of second-generation CAR T-cells (CTL019) in two CLL patients, stable cloned CAR T-cells could be continuously detected in vivo, and sustained remission for 10 years was achieved, which is encouraging [5]. For patients with chemotherapy resistance and difficulty in matching hematopoietic stem cells, CAR T-cell therapy may be the salvage regimen. Several commercial products (e.g., axicabtagene ciloleucel, lisocabtagene maraleucel, tisagenlecleucel, brexucabtagene autoleucel) have been approved by the US Food and Drug Administration (FDA) for the treatment of B-cell acute lymphoblastic leukemia (B-ALL) and other relapsed or refractory (R/R) large B-cell lymphomas (Table 1) [4,6,7].

However, this emerging cellular immunotherapy faces multiple challenges in the treatment of malignancies. On the one hand, most of the patients treated with CAR T-cells suffer from cytokine release syndrome (CRS), immune effector cell-associated neurotoxicity syndrome (ICANS), hematological toxicity, and grade 3/4 side effects that can threaten the life of patients [8,9,10]; on the other hand, short persistence of CAR T-cells in vivo is related with poor efficacy [11]. Therefore, some optimization schemes for CAR T-cells, such as multiple target treatment, discovery of new targets, and CAR construct optimization, have been widely studied to reduce adverse events and improve CAR T-cell efficacy [12,13,14,15,16]. However, the clinical application strategies of CAR T-cells, which have an essential role in the overall treatment process, have rarely been discussed. After determining that a patient is suitable for CAR T-cell therapy, it is necessary to determine the lymphatic clearance scheme and dosing strategy according to the patient’s basic condition, to help the patient obtain the best blood environment and maximize the utilization of CAR T-cells [17]. Furthermore, for refractory and relapsed hematological malignancies, the combined application of multiple treatments may lead to longer event-free survival for patients [18,19]. The implementation of combined strategies to increase the persistence or antitumor activity of CAR T-cells has become a research hotspot. In addition, the safe and effective implementation of a strategy is inseparable from the supervision of the disease and careful comprehensive management, which will promote patients obtaining complete remission (CR) and disease-free survival. In this review, we will discuss the lymphodepletion regimen, dosing strategy, combination treatment, and side effects management of clinical trials in order to provide new insights and directions for physicians and researchers (Figure 2).

**Table 1 cancers-14-04452-t001:** Pivotal clinical trials of FDA-approved CAR T-cell products.

CAR T-Cells	Clinical Trials	Malignancy	Enrolled Patients	Lymphodepletion Regimens	Dosage Regimens (CAR T Cells)	Efficacy	Side Effect (CRS, NE)
Axicabtagene ciloleucel	NCT02348216 [20]	DLBCL PMBCL TFL 2017	Adults 101/111	30 mg/m^2^ Flu and 500 mg/m^2^ Cy; On days -5, -4, and -3	2 × 10^6^ cells/kg on day 0; a single infusion	ORR 82% CR 54%	CRS 13%, NE 28%
	NCT02348216 [6]	DLBCL PMBCL TFL 2019	Adults 108/119	30 mg/m^2^ Flu and 500 mg/m^2^ Cy; On days -5, -4, and -3	2 × 10^6^ cells/kg on day 0; a single infusion	ORR 83% ORR 58% c	CRS 11%, NE 32%
	NCT03105336 [21]	I-NHL 2022	Adults 148/153	30 mg/m^2^ Flu and 500 mg/m^2^ Cy; On days -5, -4, and -3	2 × 10^6^ cells/kg on day 0; a single infusion	OR 92% CR 74%	CRS 7%, NE 19%
Lisocabtagene maraleucel	NCT02631044 [3]	r/r LBCL 2020	Adults 269/344	30 mg/m^2^ Flu and 500 mg/m^2^ Cy for 3 days;2– 7 days before CAR T cell infusion	50 × 10^6^ cells (51 patients), 100 × 10^6^ cells (177 patients), 150 × 10^6^ cells (41 patients); Including CD8+ and CD4+ CAR T cells	ORR 73% CR 53%	CRS 2%, NE 10%
Tisagenlecleucel	NCT02445248 [22]	r/r LBCL 2021	Adults 115/167	25 mg/m^2^ Flu and 250 mg/m^2^ Cy for 3 days; or bendamustine with 90 mg/m² for 2 days	0.1–6 × 10^8^ CAR T cells; a single infusion (target dose: 5 × 10^8^ CAR T cells)	OR 53% CR 39%	CRS 23%
	NCT03568461 [23]	r/r FL 2022	Adults 97/98	25 mg/m^2^ Flu and 250 mg/m^2^ Cy or bendamustine with 90 mg/m² for 2 days	0.6–6 × 10^8^ CAR T cells; a single infusion	CR 69.1% OR 86.2%	CRS 48.5%, NE 37.1%
Brexucabtagene autoleucel	NCT02614066 [24]	r/r ALL 2021	Adults 65/71	25 mg/m^2^ Flu and 900 mg/m^2^ Cy days -4, -3, and -2	1 × 10^6^ cells/kg on day 0; a single infusion	CR 56%	CRS 24%, NE 25%
	NCT02601313 [25]	r/r ML 2020	Adults 71/74	30 mg/m^2^ Flu and 500 mg/m^2^ Cy on days -5, -4, and -3	2 × 10^6^ cells/kg on day 0; a single infusion	ORR 93% CR 67%	CRS 15%, NE 31%

DLBCL = diffuse large B-cell lymphoma; CRS = cytokine release syndrome; NE = neurologic events; I-NHL = indolent non-Hodgkin lymphoma; Flu = fludarabine; Cy = cyclophosphamide; OR = overall response; r/r LBCL = relapsed or refractory large B-cell lymphomas; r/r FL = relapsed or refractory follicular lymphoma; r/rALL = relapsed/refractory acute lymphoblastic leukemia; r/r ML = relapsed or refractory mantle-cell lymphoma; PMBCL = primary mediastinal B-cell lymphoma; TFL = transformed follicular lymphoma; ORR = objective response rate; CR = complete response rate.

## 2. Lymphodepletion Regimen

After infusing into the blood, CAR T-cells specifically recognize the target antigen through the single-chain variable fragment (scFv) binding domain, which requires a suitable immunomodulatory environment for T cell survival and proliferation [26,27,28]. Lymphodepletion followed by CAR T-cell infusion remodels the immune environment for creating enough T cell pool, decreasing immunosuppressive cell populations, increasing the availability of cytokines and thus promoting lymphocyte proliferation and survival, and increasing tumor antigen presentation by inducing cell death [29,30,31]. Cyclophosphamide (Cy) and fludarabine (Flu) are two major lymphodepletion compositions, known as potent immunosuppressive agents. In one case, the overall response rate (ORR) without lymphodepletion was only 25%, while ORR rose to 58% after lymphodepletion that had made changes in cytokine profiles and had increased CAR T-cell durability [32,33]. However, how the lymphodepletion regimen impacts CAR T-cell efficacy is still unclear. Here, we discuss the influencing factors of chemotherapy pretreatment methods on CAR T-cell potency and durable remission based on completed clinical trials.

Generally, lymphodepletion with cyclophosphamide and/or fludarabine is implemented on -5, -4, and -3 days before CAR T-cell injection to create an immunosuppressive environment that provides CAR T-cell expansion in vivo [20,34]. Moreover, the chemotherapy pretreatment regimen adopted in most trials is a dose of fludarabine generally maintained at about 30 mg/m^2^·d or 25 mg/m^2^·d; the dose of cyclophosphamide is 200 to 3000 mg/m^2^·d [7,11,35,36,37], which is considered to be a relatively safe dose (Table 1). The co-administration of cyclophosphamide and fludarabine is viewed as adequate lymphodepletion. Studies have shown that a low incidence of CR and high rate of relapse may be associated with inadequate CAR T-cell persistence and expansion in vivo and cell-mediated immunosurveillance [17,38,39]. Cyclophosphamide and fludarabine-mediated lymphodepletion reshape the blood environment, prolong CAR T-cell survival time, regulate the release of activating cytokines, and improve the event-free survival time of patients [38,40,41].

Studies have shown that the effect of lymphodepletion and the reactivity of subsequent CAR T-cell therapy is dose-dependent [11]. A total of 17 patients received high-dose cyclophosphamide of 3 g/m^2^·day (HD-Cy), while 8 patients received low-dose cyclophosphamide at 1.5 g/m^2^·day (LD-Cy), with fludarabine at 25 mg/m^2^·day for 3 days. Results manifested that CAR T-cell expansion in the HD-Cy group was remarkably higher than that in the LD-Cy group. The disease response rate of the former was 94%, while the latter was only 38% [11]. This indicates that deeper lymphodepletion leads to higher clinical response due to thoroughly eliminating immunosuppression and reducing cytokine rejection of CAR T-cells. In addition, high-dose Cy increased the level of IL-7 and MCP-1 and was associated with CR and/or progression-free survival (PFS) [40,42]. However, the mechanism through which HD-Cy leads to a favorable cytokine profile has yet to be determined [11,43]. Nevertheless, in a previous study targeting diffuse large B-cell lymphoma (DLBCL), patients who received modulated chemotherapy with low a dose of cyclophosphamide (500 mg/m^2^) and fludarabine (30 mg/m^2^) had a higher level of cytokines, particularly IL-5 and IL-15, which are critical for T cell proliferative cytokines [34,44].

The addition of fludarabine to cyclophosphamide as a lymphodepleting regimen before CD19-directed CAR T-cell therapy significantly improved outcomes in patients treated with CAR T-cells [45]. Over 90% of patients who received fludarabine-containing lymphodepletion achieved CR and had no relapse, while 58% of patients without fludarabine suffered a relapse. In patients with CD19+ relapse, a loss of CAR T-cell in blood was detected by PCR analysis [38]. Incorporation of fludarabine into the lymphodepletion regimen, lower concentration of LDH, and higher platelet count before the lymphodepletion regimen containing fludarabine may lead to minimal residual disease-negative complete remission (MRD^-^ CR) [32]. Compared with the lower cumulative fludarabine exposure during lymphodepletion, patients in the higher group had an 11-month improvement in leukemia-free survival, and the CD19+ recurrence rate within 1 year decreased from 100% to 27.4% [46]. However, when the disease burden is high, the effect is weakened, so the dose of fludarabine should be formulated according to the patient’s disease burden and designed individually [47]. Interestingly, for DLBCL patients who relapsed after CAR T-cell therapy, pre-existing CAR T-cells were revitalized after the second lymphatic clearance containing fludarabine and created an anti-tumor effect with a predominant grade 2 CRS [48]. This case indicates that CAR T-cells could be reactivated under circumstances by disturbing the immune equilibrium. CAR T-cell activation and cytotoxicity depend on the immune homeostasis of the blood environment, and the blood environment after lymphatic clearance seems merely to maintain for a period time, which may be one of the reasons why patients can only achieve short-term remission. At the same time, to obtain satisfactory clinical efficacy, the degree of lymphatic clearance needs to be proportional to the patient’s disease burden; thus, a universal single dosage is not suitable [11,38]. 

## 3. Dosing Regimen of CAR T-Cells

Before lymphocytes are removed, T cells will be isolated from the peripheral blood mononuclear cells (PBMC) of patients as raw materials for subsequent manufacturing of CAR T-cells [49] (Figure 2). The preparation of CAR T-cells is about 13 days, including T cells expanding for 9 days; another study shortens the preparation time to 7 days with similar effectiveness [50,51]. Surprisingly, Gracell Bio in China has developed a fabrication technique of FasTCAR, which shortens the preparation time of CAR T-cells to 1 day [52]. FasTCAR technology transforms the activation, transduction, and expansion steps into a single “concurrent activation-transduction” step, eliminating the time for in vitro expansion of CAR T-cells [53]. After obtaining CAR T-cells and achieving the standards of identity, potency, sterility, and adventitious agents, the infusion strategy of CAR T-cells can be determined by clinicians and manufacturers based on phase I clinical trials identifying the maximally tolerated dose (MTD) [54].

In CD19-directed CAR T-cell products (axicabtagene ciloleucel, lisocabtagene maraleucel, tisagenlecleucel, brexucabtagene autoleucel) that have been proved by FDA, the single infusion dose is about 1–3 × 10^6^/kg, except that tisagenlecleucel is 0.1–6×10^8^ viable cells, of which 5×10^8^ viable cells are more suitable for DLBL and relapsed or refractory follicular lymphoma (FL) (Table 1) [3,4,6,7,20,21,23,34,55]. In adults, some CD19-targeted products in the treatment of different diseases in the dose range of 2 × 10^6^/kg showed higher ORR and CR and lower incidence of high-grade CRS and neurological event (NE) side effects. Determination of a safe/efficacious dose range is based on understanding dose-response/exposure/safety analyses. A dose-escalation test showed that 1 × 10^6^/kg is the highest single-injection tolerated dose for R/R ALL and non-Hodgkin’s lymphoma (NHL) patients; in this condition, all toxicities, including grade 4 CRS, are reversible [56]. Moreover, a lower CAR T-cell dosage of 1 × 10^5^/kg is effective and safe for treating r/r B-ALL [57]. It can be seen that the clinical results cannot intuitively find the relationship between the dose and the efficacy of CAR T-cells, except that high doses show dose-dependent toxicity. This may be due to the initial CAR T-cell viability and different immunosuppressive gene expression of the tumor microenvironment [58].

A single injection of low-dose CAR T-cells is considered safe and effective, but not with durable remission. Some researchers have studied one-time injection or segmental injection of low-dose or high-dose CAR T-cells [37,43]. A single-arm, open-label study verified the difference between these two strategies in efficacy. The trial is divided into a high-dose single infusion (5 × 10^8^/kg), low-dose single/fractionated infusion (5 × 10^7^/kg), and high-dose fractionated infusion (5 × 10^8^/kg); the numbers of patients were 6, 9, and 20, respectively [59]. The high-dose fractionated group showed the best effect, with a complete remission rate of 90% and a two-year overall survival rate of 73%, while in the high-dose single infusion group, patients suffered from refractory CRS complicated with culture-positive sepsis [59]. The low rate of adverse events of the high-dose fractionated group indicates that fractionated infusion of a high dose may alleviate immune response. Characteristics including age, disease phenotype, chemotherapy regimen received before infusion, and whether to undergo stem cell transplantation are all related to dosage selection, indicating that it is difficult to set the dosing interval to enable long-term complete remission [60]. Before receiving CAR T-cell therapy, these patients had experienced more than two lines of systemic therapy; however, at the same therapeutic dose, there were differences in the patients’ physical condition, immune homeostasis of the blood environment, disease burden, and cytokine pool after lymphatic depletion. A scoring system is set up to determine various aspects of the patient’s physical state, thereby determining the corresponding dose standard within a certain scoring range. The dosing regimen of CAR T-cells is individualized. However, studies of the clinical response of patients to CAR T-cells could provide clinicians with data for developing a personalized dosage strategy.

## 4. Combination Strategies

Anti-CD19 CAR T-cells show a strong short-term CR in the treatment of refractory and relapsed B-cell diseases. Its limitations include the inability to achieve long-term disease-free survival and risk of recurrence due to a lack of sustained expansion of CAR T-cells [58], and patients with a high tumor burden treated with allogeneic hematopoietic stem cell transplantation (allo-HSCT) could not achieve complete remission [61]. CAR T-cells or allo-HSCT achieved considerable short-term efficiency when used alone; researchers can explore combination to achieve long-term disease-free survival.

Allo-HSCT, as a kind of consolidation strategy, has shown a powerful capability for R/R B-cell lymphoma [61,62,63,64,65,66]. With a quick bridge to allo-HSCT after CAR T-cell therapy, the one-year event-free survival (EFS) was up to 73% compared with 50% for CAR T-cells alone, and the rate of severe grade 2–4 acute graft-versus-host disease (aGVHD) was less than 23.1% compared with 41.6% of allo-HSCT alone [7,67,68,69]. Moreover, the timing of allo-HSCT treatment affects patient prognosis. It was shown that the period within three months after treatment is considered to be a suitable time window for consolidation therapy [68]. MRD^-^ CR patients receiving allo-HSCT showed detectable CAR T-cells for 21 months. Jiang H et al. [69] found that allo-HSCT was an independent prognostic factor for EFS and relapse-free survival (RFS) in MRD^-^ CR patients. For patients with CD19-negative relapse, allo-HSCT may be desirable as a consolidated method because of immune reconstitution [57]. From another point of view, CAR T-cell therapy has obtained a larger range of patients for allo-HSCT [69]. In addition, CAR T-cells can be used as maintenance therapy after allo-HSCT therapy. Shi et al. found that CAR T-cells combined with lenalidomide maintenance therapy significantly prolonged PFS in patients with multiple myeloma. Lenalidomide increased the antitumor activity and persistence of CAR T-cells, which might be due to the fact that lenalidomide can maintain the long-term immune surveillance of CAR T-cells in remission, and this combined strategy promotes T cells toward the less terminally differentiated phenotype [70]. RNA sequencing and assays for transposase-accessible chromatin indicated that lenalidominde can alter T-helper response, cytokine production, T cell activation, cell-cycle control, and cytoskeletal remodeling [71,72].

CAR T-cells combined with ibrutinib, a kind of bruton’s tyrosine kinase inhibitor, improved CAR T-cells engraftment, tumor clearance, and survival in human xenograft models of resistant acute lymphocytic leukemia [73]. Ibrutinib is a crucial component of a first-line treatment option for chronic lymphocytic leukemia (CLL), and it showed superior PFS and CR in progressive CLL patients who were too frail to receive aggressive therapy [74]. Continuous use of ibrutinib leads to tolerance and tumor recurrence, and fortunately, CAR T-cells show obvious therapeutic efficacy and durability after ibrutinib failure [75]. Therefore, the combination of these two therapies may create an advance for r/r CLL patients. Compared with CLL patients treated with CAR T-cells without ibrutinib, CLL patients treated with CAR T-cells combined with ibrutinib were associated with lower CRS severity and lower serum concentrations of CRS-associated cytokines, which is consistent with pre-clinical results [76]. Moreover, one-year overall survival and PFS probabilities of a combination of CAR T-cells with ibrutinib were 86% and 59%, respectively, while anti-CD19 CAR T-cell therapy was only 38% and 50% [77]. After more than 1 year of ibrutinib therapy, the efficiency of CAR T-cell engraftment, tumor clearance, and survival were improved [73].

CAR T-cell therapy with immune checkpoint inhibitors has now moved to the clinical stage. A pre-clinical study demonstrated that CAR T-cell therapy combined with PD-1 blockade showed better tumor control, and the persistence of CAR T-cells was up to 21 days in mice [78]. Furthermore, neuroblastoma and metastatic melanoma with CAR T-cell therapy combined with PD-1 blockade indicated that PD-1 blockade can enhance cytokine production and reduce activation-induced cell death (AICD) of CAR T-cells after repeated antigen stimulation [17,79].

Moreover, the effect of the first-line chemotherapy for CLL patients can be strengthened by combining it with CAR T-cell therapy. After regular chemotherapy treatment with pentostatin, cyclophosphamide, and rituximab, patients who received CAR T-cells ranging from 3 × 10^6^ to 3 × 10^7^ CAR T-cells/kg were observed to have more than 28 months CR and modest CRS neurotoxicity [39]. In addition, the above methods combined with CAR T-cell therapy are moving to the clinical stage, and other molecular medicines such as immunomodulatory drugs, utomilumab, hypomethylating agents, Phosphoinositide 3-kinase (PI3K) inhibition, γ-secretase inhibitors, and fas blockade, are part of widely ongoing pre-clinical studies [80]. These combination strategies represent a new treatment paradigm that leverages the ability of genetically modified T cells to target and destroy tumor cells.

## 5. Occurrence and Management of Side Effects

Anti-CD19 CAR T-cell therapy led to cytokine release syndrome, hematological toxicity, and ICANS attributed to inflammatory cytokines and chemokines released by immune cells, which are the main obstacles after CAR T-cells were administrated into the blood system via intravenous injection [81]. The average onset time of CRS is 3 days after injection, and it lasts for about 7 days [82]. When the neurological event occurs, it is generally within 8 weeks and lasts for 28 days [82]. Blood events such as thrombocytopenia and neutropenia may exist throughout the whole disease event. The incidence of CRS in the milestone trial is as high as 77%, the average incidence of neurological events is more than 30%, and blood events are more than 40% [7]. Although corresponding support and drug treatments were performed during the treatment process, the final mortality rate of patients is more than 40%, accompanied by disease progression and aggravating effects of side effects [6,7,83]. Therefore, it is meaningful to review the occurrence and management of side effects. Moreover, early detection and timely control of adverse events also play important roles in the treatment of disease.

Generally, grade 1/2 CRS, mainly including fever, fatigue, nausea, mild hypoxia, or headache, is not life-threatening and is relieved after symptomatic and supportive treatments performed with close monitoring of the functional status of various organs and the level of cytokines in the blood. Grade 3/4 CRS is life-threatening to patients and is closely related to the subsequent neurotoxic events [10]. Grade 3/4 CRS is accompanied by more severe hypoxia, which requires ventilator support if necessary, and ICU care is needed in some severe cases [84]. In 2020, Pierre Sesques et al. [79,82] did a retrospective study on commercial products such as axicabtagene ciloleucel and tisagenlecleucel and found that grade 3/4 CRS and neurological events were reduced to 8% and 10% compared with the key trial in 2017, which is 11% and 32%, respectively. The lower rate of grade 3/4 side effects is due to earlier and more systematic interventions.

When CRS occurs, the levels of IL-2, IL-4, IL-6, IL-10, TNF-α, and IFN-γ in the blood are significantly increased; in particular, IL-6 is parallel to the severity of CRS. Compared with grade 1/2 CRS, grade 3/4 CRS showed a rapid increase in IL-6 in the early stage, and the patients’ fever was high and lasted for a significant period. These clinical symptoms can be used as an early prediction of severe CRS and can provide treatment strategies for adequate treatment in the later period [10,85,86]. On the other hand, the rapid expansion of infused CAR T-cells is also related to CRS [9]. At the same time, the concentration of the above cytokines also significantly increased with the expansion of CAR T-cells, which is associated with not only injection protocol but also the kind of costimulatory domain of CAR T-cells [16,36]. The IL-6 antagonist tocilizumab is the first-line treatment for CRS; however, it is usually used when supportive treatment does not work. If necessary, it can be used in combination with the immunosuppressant dexamethasone. Generally, it can provide relief after one week of treatment. However, when the tumor burden and the number of recurrences in the past are high, severe CRS requires hemodialysis or plasma exchange to overcome difficulties [10,34].

In addition, grade 3/4 CRS is related to the increased risk of subsequent infections, especially bloodstream infections that are difficult to distinguish from grade 3/4 CRS. Bloodstream infections are similar to CRS symptoms, but the treatment methods for the two are different. We can determine the infection through imaging methods and laboratory tests, such as computed tomography, bacterial examination, magnetic resonance imaging, C-reactive protein (CRP), procalcitonin, etc. When it is impossible to distinguish, clinicians should use antibiotics based on their experience [87,88]. Moreover, the incidence of disseminated intravascular coagulation (DIC) increases in patients with grade 3/4 CRS. As a result, early and proper interventions targeted to CRS-related coagulopathy greatly contribute to the control of side effects in CAR T-cell therapy [85].

Neurotoxic events occurred in the study as compared with the high incidence of CRS, but with the optimization of CAR T-cells, the incidence of CRS and neurological events decreased [37,89]. ICANS often occurs with grade 3/4 CRS. ICANS includes encephalopathy, confusion, delirium, tremor, restlessness, lethargy, and epilepsy in severe patients (Grade 3) [7]. There is no specific treatment method, other than relying on supportive treatment. Specific antibodies targeting cytokines related to neurotoxic events may be found in the future. Studies have shown that neurotoxicity is related to CAR T-cells crossing the blood–brain barrier and triggering an immune response in the brain. IFN-γ and IL-6 in the cerebrospinal fluid are nearly 40 times higher than those in the serum [90]. At the same time, the qPCR of the CAR expression indicates that it is in the cerebrospinal fluid, and the CAR copy number is three times higher [90]. Thus, the detection of components in cerebrospinal fluid can be used as a method to predict neurotoxicity [86,88,90]. Due to individual differences, CRS, hematological toxicity, and neurotoxicity may occur in patients at any time after CAR T-cell treatment. Identifying the characteristics of side effects and determining the treatment may be helpful for the positive progress of CAR T-cell therapy.

## 6. Discussion

With the clear superiority of CAR T-cell therapy in clinical trials, CAR T-cell therapy will become the standard therapy for hematological malignancies. However, the clinical application strategy of CAR T-cells has not yet achieved consensus, and there remain variability and uncontrollability. One of the preparations before CAR T-cell infusion is the preparation of the patient’s blood environment, including the important step of lymphodepletion. Currently, the lymphodepletion regimens used in commercial CAR T cell products are mainly cyclophosphamide and fludarabine. The main limitation is that the dose selection and combination strategies of lymphodepletion regimens are relatively single and do not take into account the individuality of the patient’s blood environment. Lymphodepletion, as an important cornerstone of CAR T-cell therapy, has direct anti-tumor cytotoxicity, activates innate immunity, enhances the production of homeostatic cytokines (MCP1, IL-7, and IL-15) [44], and reduces the immune regulation number of cells, such as myeloid-derived suppressor cells and regulatory T cells [91], and decreases indoleamine 2,3-dioxygenase (IDO) expression [92]. Therefore, close detection of lymphocyte numbers, amount of favorable cytokines, and clearance of immunosuppressive cells during lymphodepletion correlate with cytotoxic function and persistence after further CAR T-cell infusion and patient prognosis. At the same time, preclinical assessment of patients with these indicators can be used to identify chemotherapy drugs and combination strategies for lymphatic failure.

In CAR T-cell therapy procedures, the choice of therapeutic dose directly affects the prognosis of patients. CAR T-cells as a live cell drug have more variability than chemical drugs. In the previous discussion, we found that in pivotal experiments, the dose of each kind of CAR-T cell was the same in different diseases. However the relationship between dose and exposure is highly variable and may be affected by the design of the CAR, T-cell initial immune function, the proportion of CAR T-cell phenotype, disease burden, lymphoid depletion, and the subsequent effects of immunomodulatory therapy [54,93]. These factors determine that a single therapeutic dose cannot maximize efficacy for patients. In order to reduce side effects and improve the maximum utilization of CAR T-cells, dose escalation trials are underway [56]. Although low doses can reduce CRS, ICAMS, and hematological toxicity, CAR T-cells cannot sustainably expand, release toxic substances, and effectively eliminate tumor cells. The pharmacokinetic of CAR T-cells in patients is closely related to recurrence in patients. Therefore, the homeostasis of CAR T-cells in vivo should be adapted to the tumor cells per unit area. In future studies, studies of dose selection should incorporate multiple assessment metrics, such as lymphatic clearance, immune function acquired by CAR T-cells, structure and phenotype of CAR T-cells, disease burden, and patient underlying disease and tolerance ability [94].

Although CAR T-cell therapy is promising, limitations still exist, such as CAR T-cell exhaustion, tumor antigen escape, directional trafficking of CAR T-cells, the immunosuppressive microenvironment of the tumor, and the occurrence of high-grade side effects. In addition to directly killing tumor cells, the implementation of the combination therapy strategy can also modulate the immunosuppressive microenvironment, combat CAR T-cell exhaustion, and reduce immune side effects. Allo-HSCT is considered as an effective way to cure hematological malignancies. However, recurrence has become a major obstacle to Allo-HSCT therapy, mainly due to the loss and downregulation of autologous human leukocyte antigen (HLA), the upregulation of T cell inhibitory ligands, and the immunosuppressive microenvironment in the tumor [95]. CAR T-cells specifically recognize target antigens in an HLA-independent manner against the immune tolerance mechanism of tumor cells. At the same time, both CD8^+^ T cell subsets and CD4^+^ T cell subsets can be introduced into CAR to mediate cytolysis through granzyme/perforin independent of Fas or TNF-α signaling [96]. Preclinical studies have demonstrated that a variety of immunomodulatory drugs can enhance the efficacy of CAR T-cells from different perspectives. Lenalidomide increases the expression of nuclear transcription factor-κB (NF-κB) by phosphorylating the co-stimulatory domain CD28, reverses the decrease in filamentous actin (F-actin) polymerization in T cells, increases the cytokine IL-21 to change the metabolic pattern of CAR T-cells, and enhances the persistence and activity of the CD2 subset of CAR T-cells [97,98,99,100]. The rationale for ibrutinib binding to CAR T-cells is based on its direct effect on T cells and its ability to destroy the immunosuppressive TME. Ibrutinib can reduce the expression of CXC chemokine receptor 4 (CXCR4) in leukemia cells, and inhibit the signal transduction downstream of CXCR4, disrupt the homing of leukemia cells to lymphoid tissue, and promote its release into the circulation, reducing malignant CD200 (OX2) expression on cells and decreasing expression of immunosuppressive cytokines [101]. PD-1 blocking antibody increases cytokine production and restores CAR T-cell activity in vitro and in vivo [102]. Decitabine increases the activity of CD19-targeted CAR T-cells by increasing the immunogenicity of tumor cells and increases the transcription and cell surface expression of CD19 in lymphoma cell lines [103]. Preclinical studies have provided substantial evidence that combination therapy can overcome the limitations of CAR T-cells themselves and the inhibitory effect of the tumor immune microenvironment, enabling adoptive cell therapy to achieve sustained expansion and cytotoxicity. More clinical trials should be designed to determine the certainty of the efficacy of combination therapy; the development of sustainable combination therapy strategies is a direction for future research.

The success of CAR T-cell therapy lies in the timely and effective management of side effects and the ability to identify the link between the test indicators and the occurrence of side effects. In adoptive cell therapy, CRS is the most common and serious complication; it is a cytokine cascade that occurs with CAR T-cell activation. The temporal peak of this cytokine release is influenced by the co-stimulatory domain, and CRS in CAR T-cells containing CD28 structures appears early and at high levels. Cascade release of cytokines can also trigger the cytokine release of other host antigen-presenting cells and other T cells, while monocytes, macrophages, and dendritic cells are major sources of IL-6, a key player in CRS [104]. Excessive inflammation activates the vascular endothelium, increases levels of vascular permeability factors, and decreases levels of the endothelial stabilizer Ang-1, leading to loss of vascular integrity, hemodynamic instability, capillary leakage, and consumptive coagulopathy in CRS, and then triggering a series of systemic side effects [105]. Additionally, pediatric patients with B-ALL with higher tumor burden had greater CAR T-cell expansion and higher CRS severity [106]. The mechanism of CRS occurrence suggests events that need to be monitored in the management of side effects in clinical studies. The occurrence of ICANS is often secondary to severe CRS, and there is no direct relationship between the occurrence of ICANS and CAR T-cells. Preclinical studies have shown that IL-1, IL-6, and granulocyte macrophage colony stimulating factor (GMCSF) are the most important factors for the development of ICAN after CAR T-cell therapy [107]. In addition, infection events during treatment need to be closely monitored. The occurrence of infection is closely related to the patient’s blood immunodeficiency. The reasons include tumor cell suppression of normal immune cells, lymphatic depletion, and subsequent systemic reactions to serious complications [108]. The mechanism of these side effects is variable and complex, and it may require a professional team devoted to management of side effects, making it an important means to promote efficacy.

## 7. Conclusions

In this review, we discussed the clinical application strategies of CAR T-cells, which play a crucial role in the process of CAR T-cell therapy (Figure 2). Lymphodepletion regimen, dosing strategy, and side effect management are indispensable steps for CAR T-cell therapy. These processes should be taken into careful consideration before and during treatment to optimize the efficacy and reduce the severity of side effects. Furthermore, the development of combination therapies helps maximize CAR T-cell function, which is increasingly studied. CAR T-cells combined with allo-HSCT is a promising strategy that may cure hematological malignancy by overcoming tumor escape mechanisms and improving anti-tumor activity and persistence. In addition, the efficacy and safety of CAR T-cell application depends on the prediction and management of side effects. The monitoring of CRS and ICANS can predict the prognosis of patients for earlier intervention, especially monitoring changes in cytokines. In the future, CAR T-cells will be widely applied to cancer, including hematologic malignancy and solid tumors, due to the high efficiency of the anti-tumor effects. The exact CAR T-cell efficacy is based on the patient’s underlying disease, disease burden, and immune environment, which are all uncontrollable factors. However, when the patient is in the treatment program, ensuring that the patient obtains the longest complete remission time and disease-free survival depends on the coordination of clinical measures. As we mentioned previously, these factors are relatively controllable. More clinical trials should be investigated to explore the effect of lymphatic depletion and dose on peak CAR T-cell expansion, tumor killing, and the occurrence of side effects, which may relate to product type selection, disease burden, and modulation of the immune environment. In many ways, this can help clinicians and researchers optimize CAR T-cell therapy, expand the therapeutic window, and improve the availability of this emerging cancer immunotherapy.

## Figures and Tables

**Figure 1 cancers-14-04452-f001:**
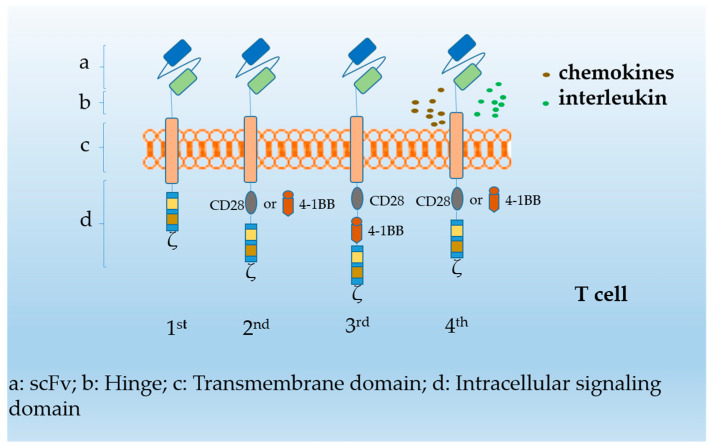
Molecular structure of CAR in four generations of CAR T cells. The CAR structure is embedded in T cells by genetic engineering and is mainly composed of a single-chain variable fragment (scFv), a transmembrane domain, a costimulatory domain (CD28 or 4-1BB), and an intracellular activation domain CD3ζ composition. CAR T-cells utilize the specific recognition of extracellular structures and the cytotoxicity of T cells to kill target tumor cells, successfully avoiding the restrictive effect of major histocompatibility complex (MHC) class I and improving the lethality of tumor cells. The first-generation CAR T-cell scFv couples with intracellular CD3ζ to obtain specific antigen recognition ability, and the second-generation CAR T-cells add the CD28 or 4-1BB costimulatory domain structure to overcome the persistent poor activation of T cells, low activation intensity, and cytokine secretion disorder. The third-generation CAR T-cells that combine both CD28 and 4-1BB costimulatory domains have improved cytotoxicity and persistence, while fourth-generation CAR T-cells with one costimulatory domain are genetically engineered to introduce specific genes to perform cytotoxic functions and maintain T cell activation.

**Figure 2 cancers-14-04452-f002:**
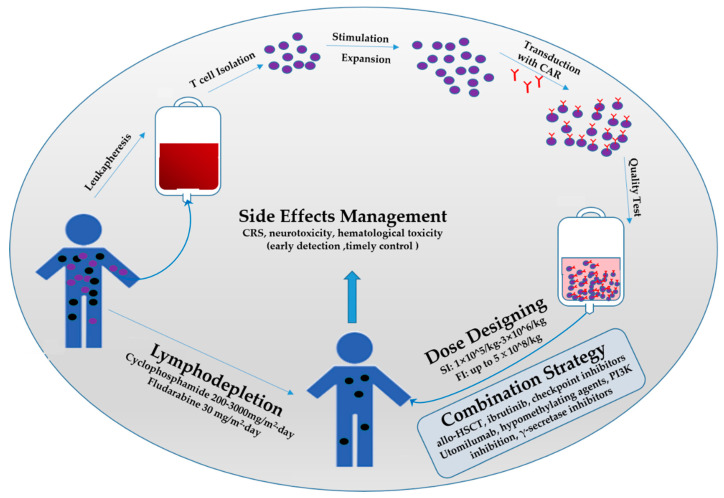
Process and clinical application strategies of CAR T-cell therapy. The preparation of CAR T-cells contains leukapheresis, T cell isolation, and stimulation and expansion in vitro; then, T cells are transduced with the CAR structure targeting a specific antigen. Then, the qualified CAR T-cells are infused into the patient via intravenous injection. In the clinical stage, lymphodepletion, dose designing of CAR T-cells and side effects management must be seriously considered during clinical treatment. If necessary, combination strategies should be taken into consideration. SI: Single Infusion; FI: Fractionated Infusion.

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
