# Peer review of "Clinical Strategies for Enhancing the Efficacy of CAR T-Cell Therapy for Hematological Malignancies"

_cancers, 2022, doi:10.3390/cancers14184452_

Round 1
Reviewer 1 Report
I have no major concerns or criticisms for this well written manuscript.
Suggestions -
- Authors appropriately titled the manuscript as 'Clinical Strategies", but should include 'hematological malignancies' CAR T-cell therapy as the article is solely focused on hem malignancies.
Author Response
We really appreciate your consideration of our manuscript and all your help with the review process. Here we summarize a list of point-to-point responses to your comments.
1.Authors appropriately titled the manuscript as 'Clinical Strategies", but should include 'hematological malignancies' CAR T-cell therapy as the article is solely focused on hem malignancies.
Answer: Thank you for your comments on our manuscript. I have changed the title to “Clinical Strategies for Enhancing the Efficacy of CAR T-cell Therapy for Hematological Malignancies” that is more in line with this manuscript.

Reviewer 2 Report
General comment:
The efficacy of CAR T cells as a potential cell therapy in the treatment of hematological tumors is obvious to all. Authors discussed the clinical application strategies of CAR T-cells, which play a crucial role in the process of CAR T-cells therapy, including lymphodepletion regimen, dosing strategy, combination strategy and side effects management. These processes should be taken into careful consideration before and during treatment to optimize the efficacy and reduce the severity of side effects.
This report is considered as important information for physicians. If the authors could add information as below, this paper would be more informative.
Specific comment:
1. Shi et al. insist that lenalidomide treatment after CAR-T therapy was able to improve CAR-T cell maintenance. Please add this information in the “Combination Strategy”. (Shi X, Yan L, Shang J, et al. Anti-CD19 and anti-BCMA CAR T cell therapy followed by lenalidomide maintenance after autologous stem-cell transplantation for high-risk newly diagnosed multiple myeloma. Am J Hematol. 2022;97:537-547. doi: 10.1002/ajh.26486.)
2. The characteristics of long-term persisting CAR T cells have not been extensively studied. Melenhorst JJ. et al. reported two cases in which CART cells were maintained for more than 9 years. The information in this paper is very important, so please explain it in the “Introduction” or elsewhere. (Melenhorst JJ, Chen GM, Wang M, et al. Decade-long leukaemia remissions with persistence of CD4+ CAR T cells. Nature. 2022 ;602:503-509. doi: 10.1038/s41586-021-04390-6. )
Author Response
We also greatly appreciate the reviewers’ time and effort in providing constructive comments and suggestions on our manuscript. Here we summarize a list of point-to-point responses to your comments.
1. Shi et al. insist that lenalidomide treatment after CAR-T therapy was able to improve CAR-T cell maintenance. Please add this information in the “Combination Strategy”. (Shi X, Yan L, Shang J, et al. Anti-CD19 and anti-BCMA CAR T cell therapy followed by lenalidomide maintenance after autologous stem-cell transplantation for high-risk newly diagnosed multiple myeloma. Am J Hematol. 2022;97:537-547. doi: 10.1002/ajh.26486.)
2.The characteristics of long-term persisting CAR T cells have not been extensively studied. Melenhorst JJ. et al. reported two cases in which CART cells were maintained for more than 9 years. The information in this paper is very important, so please explain it in the “Introduction” or elsewhere. (Melenhorst JJ, Chen GM, Wang M, et al. Decade-long leukaemia remissions with persistence of CD4+ CAR T cells. Nature. 2022 ;602:503-509. doi: 10.1038/s41586-021-04390-6. )
Answer: Thank you for your professional advice on our manuscript. Based on your proposal, I added the two parts you mentioned to the introduction and the combination therapy strategy, respectively. Added content is marked in red in the text.

Reviewer 3 Report
This is a very interesting review, relevant to the field, that goes thought the management of the clinical limitations associated with CAR-T cell therapies, during the lymphodepletion and dosing regimen, the combination strategies, etc. It reviews the clinical strategies for enhancing the efficacy of CAR-T cell therapy and overcome the described limitations. The manuscript is well written.
Revisions:
· The quality of the figures must be improved, especially for Figure 1. The used font should be the same as in the rest of the manuscript for both figures.
· CAR-T cell molecular structure development could be described in further detail in the introduction or in an independent section to accompany the Figure 1.
· It would be interesting if the molecular immune mechanisms behind the limitations in each section of CAR-T cell therapies could be further explained.
· Also, a very interested reader would appreciate a section and discussion of other strategies to improve CAR-T cell therapies from its preclinical development (novel tumour antigen screening, improvement of the CAR intracellular structure, how to overcome the tumour microenvironment barrier for the use of CAR-T cells for solid tumours…).
· It would be interesting if the clinical trials developing CAR-T cell therapies with the aims of overcoming the described clinical limitations and improving the efficacy and safety of CAR-T could be discussed and detailed in a table. The future preclinical and clinical perspectives of adoptive cell therapies and CAR-T cells could be further discussed.
· The content of the “discussion” section is OK, but I would name it as a “conclusion” section.
Author Response
We greatly appreciate the reviewers’ time and effort in providing constructive comments and suggestions on our manuscript. Here we summarize a list of point-to-point responses to your comments.
1.The quality of the figures must be improved, especially for Figure 1. The used font should be the same as in the rest of the manuscript for both figures.
Answer: Thank you for your comments. The font of all tables and figures have been the same of in the rest of the manuscript.
2. CAR-T cell molecular structure development could be described in further detail in the introduction or in an independent section to accompany the Figure
Answer: Thank you for your comments. I have added more detail description about the four generations of CAR T-cells in the legend of Figure 1.
3. It would be interesting if the molecular immune mechanisms behind the limitations in each section of CAR-T cell therapies could be further explained.
Answer: Thank you for your comments. The purpose of this article is to discuss strategies for clinical application, a discussion of its mechanisms may have been overlooked. In the revised manuscript, I have added a discussion section, which mainly deals with the mechanisms related to the application strategy. However, due to the large number of molecular mechanisms involved in each part of the application strategy, it is not discussed in detail in this review.
4. Also, a very interested reader would appreciate a section and discussion of other strategies to improve CAR-T cell therapies from its preclinical development (novel tumour antigen screening, improvement of the CAR intracellular structure, how to overcome the tumour microenvironment barrier for the use of CAR-T cells for solid tumours…).
Answer: Thank you for your comments. Our manuscript focuses on the relevant strategies of CAR T cells in clinical treatment, including lymphodepletion regimen, dosing strategy, combination treatment, and side effects management. We believe that clinical evaluation should be an important consideration in the development of CAR T-cell. These aspects may be mentioned in some articles in the field, but the article summarizing and discussing these four aspects has not been seen. Discussing the clinical events involved in the application of CAR T cells could provide some different insights into CAR T cells in clinical progress. The content you mentioned is also of interest to our research group. This part of the relevant content was written by Zengping Liu, the second author of this manuscript, and has been published in the Chinese Journal of Oncology. If you are interested, you can refer to this review titled Improving the effectiveness of CAR-T cell immunotherapy and A new strategy for security.
5. It would be interesting if the clinical trials developing CAR-T cell therapies with the aims of overcoming the described clinical limitations and improving the efficacy and safety of CAR-T could be discussed and detailed in a table. The future preclinical and clinical perspectives of adoptive cell therapies and CAR-T cells could be further discussed.
Answer: Thank you for your comments. In the revised manuscript, I have added a discussion section, which mainly deals with the mechanisms related to the application strategy, and discussed the development direction and prospect of future CAR T cell therapy with respect to clinical strategies.
6. The content of the “discussion” section is OK, but I would name it as a “conclusion” section.
Answer: Thanks for your valuable opinion. I very much agree with your comments, so I have added a discussion section and changed the original discussion to a conclusion section.
